# Precision and Personalized Medicine: How Genomic Approach Improves the Management of Cardiovascular and Neurodegenerative Disease

**DOI:** 10.3390/genes11070747

**Published:** 2020-07-06

**Authors:** Oriana Strianese, Francesca Rizzo, Michele Ciccarelli, Gennaro Galasso, Ylenia D’Agostino, Annamaria Salvati, Carmine Del Giudice, Paola Tesorio, Maria Rosaria Rusciano

**Affiliations:** 1Clinical Research and Innovation, Clinica Montevergine S.p.A., 83013 Mercogliano, Italy; ostrianese@gmail.com (O.S.); c.delgiudice@clinicamontevergine.it (C.D.G.); 2Laboratory of Molecular Medicine and Genomics, Department of Medicine, Surgery and Dentistry, Scuola Medica Salernitana, University of Salerno, 84084 Baronissi, Italy; frizzo@unisa.it (F.R.); yleniadag87@gmail.com (Y.D.); asalvati@unisa.it (A.S.); 3Department of Medicine, Surgery and Dentistry, Scuola Medica Salernitana, University of Salerno, 84084 Baronissi, Italy; mciccarelli@unisa.it (M.C.); ggalasso@unisa.it (G.G.); 4Unit of Cardiology, Clinica Montevergine S.p.A., 83013 Mercogliano, Italy; paolatesorio@gmail.com

**Keywords:** precision medicine, personalized medicine, clinical application, genomics

## Abstract

Life expectancy has gradually grown over the last century. This has deeply affected healthcare costs, since the growth of an aging population is correlated to the increasing burden of chronic diseases. This represents the interesting challenge of how to manage patients with chronic diseases in order to improve health care budgets. Effective primary prevention could represent a promising route. To this end, precision, together with personalized medicine, are useful instruments in order to investigate pathological processes before the appearance of clinical symptoms and to guide physicians to choose a targeted therapy to manage the patient. Cardiovascular and neurodegenerative diseases represent suitable models for taking full advantage of precision medicine technologies applied to all stages of disease development. The availability of high technology incorporating artificial intelligence and advancement progress made in the field of biomedical research have been substantial to understand how genes, epigenetic modifications, aging, nutrition, drugs, microbiome and other environmental factors can impact health and chronic disorders. The aim of the present review is to address how precision and personalized medicine can bring greater clarity to the clinical and biological complexity of these types of disorders associated with high mortality, involving tremendous health care costs, by describing in detail the methods that can be applied. This might offer precious tools for preventive strategies and possible clues on the evolution of the disease and could help in predicting morbidity, mortality and detecting chronic disease indicators much earlier in the disease course. This, of course, will have a major effect on both improving the quality of care and quality of life of the patients and reducing time efforts and healthcare costs.

## 1. Introduction

Over the last century, there has been a gradual increase in average life expectancy [1]. 

Today, for the first time in history, most people can expect to live into their sixties and beyond. Population aging represents one of the most important demographic features worldwide [2] and it is poised to become one of the most significant social transformations of the twenty-first century, with implications in all aspects of society [3,4]. One of the major consequences of a rapidly aging population [5] is the increasing burden of chronic diseases associated with age and related healthcare costs [6]. The most common chronic diseases linked to an aging society are the following: cardiovascular disease; diabetes; neurodegenerative diseases; most cancers [7]. The management of complex chronic diseases is becoming a serious social and economic trouble for the world’s elderly population and represents a huge challenge for national and international health care budgets [8]. 

The general consensus is that these types of disorders can be best tackled by effective primary prevention, demonstrating that pathological processes begin years before the appearance of clinical symptoms. Effective primary prevention can be challenging. The growing investment in genomic research has raised great expectations concerning its effects on biomedicine to study susceptibility to cancer and other chronic diseases and to promote new preventive interventions [9]. An ambitious challenge for medicine is to guarantee targeted care paths beginning with more personalized approaches. To achieve this goal, in the genomic era we have already approached an exciting period of medicine, where a convergence of genomics, bioinformatics and new molecular techniques, promises to improve our understanding of the genetic basis of many diseases, as shown in Figure 1. With this greater understanding comes the possibility of redefining disease mechanisms at higher resolutions and, along with this, targeting with more precise therapy [10] (e.g., preventing its onset and enabling early detection, as well as tailoring therapy to patients’ characteristics [11]). 

Among age-related degenerative conditions, cardiovascular and neurodegenerative diseases account for the majority of hospitalizations, healthcare spending and mortality worldwide. Due to the large number of people who are potentially affected by these conditions, this puts pressure on health systems, increasing the demand for care, services and technologies to prevent and treat these types of non-communicable diseases [12]. 

What these conditions have in common is their relatively slow development, occurring over the course of many decades. Due to this, they need a long-term and complex response, coordinated by different health professionals with access to the necessary drugs and equipment [13], but, at the same time, this could provide ample opportunities to identify those at risk, offer preventative strategies, and start therapies early in the course of the disease mechanism [14].

To this purpose, the present review has been thought to discuss the main components, which may be part of precise and specific individual treatment programs applied to neurodegenerative and cardiovascular disorders. We focus on this new approach, with precision medicine defined as “an emerging approach for disease treatment and prevention that takes into account individual variability in genes, environment, and lifestyle for each person” [15]. 

The application of precision medicine to cardiovascular diseases, as in many other diseases linked to high mortality risk in the population, holds the promise of improving health as well as revolutionizing prevention and treatment options, similar to what has occurred in the field of oncology. The clinical translation and application of precision medicine can be achieved by developing the personalized treatment of patients based on the premise that prevention is better than treating [16]. Furthermore, we also want to outline the difference between personalized medicine (PeM) and precision medicine (PM) into managing these complex chronic disorders by demonstrating the potential benefits of precision medicine novel technologies demonstrated to date in order to offer insights into routine clinical practice.

## 2. What Is the Difference between Personalized Medicine and Precision Medicine

### Precision vs. Personalized Medicine: What Is the Difference?

To better understand how precision medicine can be applied to personalized treatments for the management of patients with cardiovascular and neurodegenerative disorders, it is first necessary to clarify the difference between personalized medicine and precision medicine. A minimum common denominator characterizes biomedical research today—finding causes, focusing on the individual and clarifying the details. This has been defined as the personalized medicine (PeM) approach, and more recently as precision medicine (PM). 

According to the National Research Council, “personalized medicine” is an older term with a closer meaning to “precision medicine” [10]. It first appeared in published works in 1999; however some of the field’s core concepts have been in existence since the early 1960s [17]. 

Although it is not uncommon that the two terms are often used interchangeably, there is a conceptual distinction between personalized medicine and precision medicine that refers to a different approach to patients [10].

Hippocrates introduced a personalized medicine approach for patient care thousands of years ago [17]. He wrote about the individuality of disease and the necessity of giving “different drugs to different patients, for the sweet ones do not benefit everyone, nor do the astringent ones, nor are all the patients able to drink the same things” [17,18,19]. 

In general, it is accepted that personalized medicine is a medical treatment to improve the delivery of drug therapy or preventive care, tailored to the needs of individual patients [20]. It refers to an approach involving the use of an individual’s genetic and epigenetic information [21], with particular attention to their preferences, beliefs, attitudes, knowledge and social contexts. 

This approach is based on the use of new technologies and relies on individuals’ unique molecular profiles and plays an important role in response to what makes them vulnerable to certain diseases [22].

Together with research and clinical care progress, the policy enabling personalized medicine has changed the medical approach and consequently transformed the health care system too [23]. The creation of collaborative networks, among medical centers and highly qualified specialists who are equipped with more precise tools, can develop therapy protocols that are targeted to patient groups that do not respond to medications as intended and for whom the traditional health systems have otherwise failed [24]. 

This has led to an improvement in our ability to predict which medical treatments will be safe and effective for each patient, and which ones will not be because it may not only minimize harmful side effects and ensure a more successful outcome, but can also help contain costs compared with a “trial-and-error” approach to disease treatment [24]. 

Until now, one major limitation of standard medical treatment is that most of them have been designed for the “average patient” [20], assuming that all patients with the same symptoms of disease share a common patho-phenotype and, therefore, should be treated similarly [25]. As a result of this “one-size-fits-all” approach [26], treatments can be very successful for some patients but not for others, as shown in Figure 2. 

Although the term precision medicine has become very popular only recently, enabled by scientific as well as political initiatives [27], the concept has been part of health care for many years. 

The President’s Council of Advisors on Science and Technology, Executive Office of the President of the United States, with a report in September 2008 [10], has specified that the term Personalized Medicine “does not literally mean the creation of drugs or medical devices that are unique to a patient, but rather the ability to classify individuals into subpopulations that differ in their susceptibility to a particular disease or their response to a specific treatment [28] 

In January 2015, when President Barak Obama launched the Precision Medicine Initiative [29], it was the first time that the general population was exposed to the greater concept of precision medicine [14,30] as a bold new effort to revolutionize how to improve, health care and “help people live longer, happier, and healthier lives.” 

In “Precision medicine” the focus is on identifying optimal care based on a unique personal profile (i.e., individual differences in genetics, exposures, lifestyle and health factors) in order to determine disease susceptibility, understand the clinical course of the disease and prescribe appropriate drugs or other therapies in defined subpopulations of patients, rather than on the average population [25]. 

The expansion of precision medicine is based on using multiple sources from genomics, biological data, transcriptomics and proteomics crucial for prediction, in order to be more precise and accurate in diagnoses, definitions and treatments of disease subtypes [31]. 

Therefore, precision medicine is a new medical strategy that defines a disease at a higher resolution to enable the more precise targeting of subgroups of disease with new therapies; prominent examples include cystic fibrosis and cancer [32]. 

The discovery of biomarkers that can be identified before the emergence of overt clinical symptoms, together with technological development, have paved the way for rapid genomic discovery, thereby transforming the current healthcare approach from one centered on precision medicine to a more comprehensive focus on precision health [33], offering the possibility of preventing disease altogether [34]. 

The future of precision medicine will enable health care providers to tailor treatment and prevention strategies to people’s unique characteristics, including their genome sequence, microbiome composition, health history, lifestyle and diet. 

## 3. Brief View of Sequencing Eras

Historically, the concept of the DNA sequencing method was introduced in 1975 by Sanger, opening a new paradigm of research and medical genomics. For many years, the Sanger method has been the principal method used among the sequencing procedures, also playing a key role in the completion of the Human Genome Project, which lasted 13 years and ended in 2003 [35,36]. Nowadays, this technology is still a gold standard approach used in diagnostic procedures of single-gene testing, but the limitations of high cost, reduced sensitivity and time demand have determined the passage from low- to high-throughput sequencing, starting Second Generation Sequencing (2ndGS), also known as Next Generation Sequencing (NGS) or the massively parallel sequencing era [37].

NGS transformed our ability to study the human genome or any organism of interest thanks to the reduction in costs and the possibility of obtaining thousands or millions of DNA sequences in a high-throughput and efficient manner and producing unachievable results with the traditional Sanger procedure [37]. The main companies that participated in the development of NGS technologies were Illumina, Roche 454 and Ion Torrent (Thermo Fisher, Waltham, MA, USA) through two different methods based on Sequencing by Synthesis (SBS). Roche 454 and Ion Torrent approaches provides the clonal amplification of DNA fragments ligated on beads by “emulsion PCR”, generating thousands of copies of the same molecules [38]. Instead, the solid-phase amplification by Illumina is based on oligonucleotide anchorage, previously amplified by PCR on the surface of a flow cell, starting cluster generation by the “bridge amplification” process [39]. 

In the last few years, MGI Tech Co., Ltd. (MGI) became a new competitor producer of high-throughput sequencers. MGI technologies are based on Combinatorial Probe-Anchor Ligation (cPAL) [40] sequencing, based on rolling circle DNA amplification (RCA), forming a final structure of concatemer sequences, called DNA nanoballs (DNBs). 

The advances in sequencing technologies had a strong impact on genetic studies, improving the knowledge of genetic variations of human genomes and their effects on health and disease. This new approach in the study of genetic disease has elicited a revolution in the understanding of the complex biological networks associated with multifactorial and polygenic diseases, such as diabetes, cancer and cardiovascular and neurological disorders [41,42,43]. All of that determined a rapid advancement in the field of genomics and biological research and medical diagnostics, allowing for the translation of these technologies in a “bench to bedside” manner. The individual genome profile is one of the key components of precision medicine and it can be used to customize diagnosis and predict therapy patient responsiveness.

## 4. NGS Applications in Precision Medicine

The benefits of NGS technologies in the diagnostic procedure is the speed, accuracy and increasing reliability of data, even if the difficulty of managing and interpreting a large amount of information is still an unresolved problem in genotype–phenotype correlation. NGS includes different applications, such as Whole Genome Sequencing (WGS), Whole Exome Sequencing (WES) and Targeted Sequencing capable of identifying variants from different DNA sources, such as body fluids (e.g., blood, saliva, cerebrospinal fluid) or tissue biopsy, as shown in Table 1. Moreover, NGS-based detection methods allow for the identification of rare and low-frequency variants on a massive scale and enable higher analytical sensitivity in samples with low input DNA, a particularly important challenge in the case of poor availability of starting material, such as tissue or biologic fluid [44].

WGS currently represents the most comprehensive strategy with the potential to identify every genetic variation that plays a role in human disease. The data generated are extended to the entire genome (~3 billion bases), including coding and non-coding regions, providing additional information on mutations in regulatory regions, such as promoters, UTRs, enhancer elements and chromatin marks, as well as the solid identification of large structural variants, copy number variants (CNVs), gene fusions, inversions and deletions (indel) [45]. Usually, in WGS experiments about 3–4 million variants per individual are identified [46,47], although currently it is still very difficult to interpret the variants outside the protein-coding regions.

Although WGS provides a more exhaustive point of view of the genome variations, target-enrichment strategies offer a suitable alternative, balancing cost and benefit and becoming, in the last few years, a popular method for identifying many driving genes and pathways involved in genetic disease, such as congenital heart and neurodegenerative diseases [48,49]. Targeted sequencing allows for the capturing of specific genomic regions from a DNA sample before sequencing, which increases coverage, facilitates the multiplexing of samples and simplifies analysis. These procedures are based on two different selection strategies: amplicon-based assays and hybridization capture.

The multiplex PCR is an amplicon-based target enrichment reaction, where multiple primers directed to specific and targeted regions are used, allowing for multiple long-range PCRs in parallel in one PCR reaction tube [50]. Some limitations of this method, given to the possibility to produce unspecific amplicon or primer interference, has been recently overcome by RainDance Technologies (Hercules, CA, USA), through the introduction of Digital PCR in droplets (ddPCR). This method generates millions of microdroplets containing a single primer pair that allow a single PCR reaction [51], but has some limitations, such as the impossibility of evaluating the presence of multiple mutations in the same amplicon. An alternative target procedure includes the use of molecular Inversion Probes (MIP), a single-stranded DNA able to anneal the flanked region of the target DNA. Once the DNA polymerase fills the gap between target and MIP sequence, the amplicon is ligated and circularized, allowing the sequencing of only the circularized fragments [52]. 

Hybridization capture strategies are the basis of target enrichment technologies, such as SureSelect and HaloPlex (Agilent Technologies, Santa Clara, CA, USA), Nextera (Illumina, San Diego, CA, USA), and SeqCap (Roche, Basilea, Switzerland) [53]. These capture methods use DNA or RNA 5′ biotin-modified oligonucleotide probes (baits) in solution, which selectively hybridize target regions in the DNA sample, previously fragmented. The magnetic streptavidin beads allow for the binding of the biotinylated probes and the selected regions are enriched by PCR reaction before sequencing [54].

The target enrichment strategy, based on Array Hybrid Capture, was originally used for sequencing the entire human exome. Nowadays, WES sequencing is mainly performed by in-solution hybridization capture because it is more rapid and convenient than PCR and, therefore, performs better with large quantities of data belonging to the exome.

WES allows for the focusing of the sequencing on the protein-coding regions that constitute 2% of the human genome (∼20,000 genes) but it should contain approximately 85% of the disease-associated mutations that affect the function of individual proteins [55]. WES data typically identify ∼25,000 coding variants per individual [56]. This technique also has some limitations, due to the protein not annotated in the human genome and the impossibility to detect structural variants and potentially non-coding element variations. Despite these limitations, WES is widely used in research and clinical application because it allows the parallel screening of a large number of samples, offering rapid and reliable methods used in prenatal diagnosis, in the detection of driving mutations causing heterogeneous disorders, in screening procedures and in the management of patient treatments [57]. 

The evolution of targeted NGS sequencing allows these technologies to become a fundamental part of clinical practice, thanks to the implementation of hotspot or multigene customized panels, which overcome the elevated cost problem, reduce the risk of incidental findings, and delete all the unused sequence information, as with the non-annotated protein-coding regions. Nowadays, many panels are available that collect well-studied genes commonly involved in several diseases. These panels are used for diagnostic/prognostic procedures, pharmacogenomics, the screening of associated mutation diseases and the selection of molecular target drugs for individual therapies [58]. As an example, NGS-based gene panel-based tests are among the main choices for diseased-patient medical investigations in daily practice. In this context, one of the more comprehensive panels—the TruSight Cardio Panel—is designed to measure the genetic profiling of 174 genes and validate relations to 17 different inherited cardiac conditions, such as cardiomyopathies, arrhythmias and aortopathies [59]. Over the last few years, greater attention and investment in sequencing technology development led to the advent of third-generation methods and, consequently, the beginning of Third Generation Sequencing (3rdGS), based on single DNA molecule sequencing and the ability to detect the nucleotide sequence in real-time, overcoming the PCR amplification step required in NGS technologies and decreasing the error insert in amplified DNA fragments [60]. Moreover, 3rdGS is characterized by the generation of long reads by high processive DNA polymerase, increasing genome alignment resolution, especially in repetitive regions. The first (3rdGS) commercialized platform, proposed by Pacific Biosciences, uses the Single-molecule real-time (SMRT) sequencing [61], based on the fluorescence detection of labeled nucleotides during the replication process. The nucleotide incorporation occurs in wells, called zeromode waveguides (ZMWs), containing a single-strand DNA template and DNA polymerase, the latter directly fixed at the bottom of the well. Once emitted light is recorded, the polymerase cleaves the dNTP-bound fluorophores, which was turned away, allowing for the addition of the next labeled dNTP. 

Recently, Oxford Nanopore Technologies(Oxfors, UK) (ONT) introduced nanopore technologies in TGS by the development of MinIOn platform. Nanopore sequencing directly detects current variation generated by native DNA or RNA passing in the hole of nanopore membrane immersed in an electrolyte solution, when a potential difference has previously been applied. The detection of the feature disruption of current, due to the passage of several combinations of nucleotide bases, makes it possible to identify the sequence of analyzed molecules.

The improvements associated with 3rdGS technologies have led to advances in read quality but significant advances are required to overcome some limitations, such as fresh starting material or intact cells and, moreover, issues associated with high sequencing error rate

Long-read sequencing technologies are particularly efficient in de novo genome assembly [62] or complex structural genomic sequencing. SMRT, for example, allowed the sequencing of long stretches of CGG as a Mycobacterium tuberculosis genome, characterized by high redundancy (∼65.6% of GC bases) or short tandem repeats (STPs) implicated in X fragile syndrome (FXS) [63,64]. 

The feasibility and versatility of the NGS approach make this technology suitable for fields of clinical and diagnostic applications, which tend to the development of precision medicine. Nevertheless, there are still many challenges to overcome by NGS to become a standard investigation procedure in patient diagnosis and therapeutic investigations, such as the management and interpretation of the enormous amount of data produced and the necessary implementations of the results, the reduction in the hands-on time for sample preparation and the development of user-friendly analysis software.

### 4.1. Neurodegenerative Diseases

Neurodegenerative diseases (NDs) are a broad group of chronic disorders that lead to a progressive loss of neuronal functions and gradually compromise the normal activities of the human brain. Patients affected by these types of disorders display variable clinical features, including memory loss, speech difficulties and motor impairment [65]. 

It has been estimated that such brain disorders have afflicted almost 7 million people over 65 years in Europe, but considering the general aging of the human population, the number of cases is expected to increase even more in the next year [66,67].

Usually, neurodegenerative processes start some years before the appearance of clinical symptoms, after which patients need special therapies to face the disabilities. As a consequence, it is necessary to counteract the pathological mechanisms as soon as possible, before the manifestation of clinical evidence [68]. 

Recently biomedical research has made many steps forward and all of the mechanisms at the base of the pathologies are still not well deciphered and a unique therapy does not seem to be the winning approach to solve the problem.

Indeed, many neurodegenerative diseases, such as Alzheimer’s disease (AD) and Parkinson’s disease (PD), are clinically heterogeneous disorders with a strong genetic component that can present variable clinical courses, which prevent the application of the same treatment to all patients, even when they have the same pathology [69,70,71]. In particular, these subgroups present various molecular characteristics, so the goal of researchers and clinicians now is to discover and catalogue all these variants. The ultimate objective is then to be able to understand the molecular causes of neurodegenerative diseases and to find effective targeted therapies. In short, it is necessary to discover biomarkers that, similar to biological labels, allow for identifying and cataloguing each subtype. Once this result is obtained, it will be possible to understand, with certainty, which sub-variant the individual patient is affected by and make the treatment personalized, modelling it on the basis of this information [34]. 

Another important aspect to consider is that not only the genetic component, but also epigenetic modifiers and non-genetic factors, such as nutrition, smoking habits, physical exercise, family history, chemical exposure, microbiome or the concomitant presence of other pathologies, can influence the dynamic of neurodegenerative evolution [72]. 

In recent years, the application of new technologies has provided a more complete view with respect to the complexity of these pathologies, highlighting the presence of different susceptibility factors with a specific impact on the development and progression of the disorder, hence encouraging the development of a precision medicine approach [73]. 

In this context, NGS technologies, by allowing for collecting a huge amount of digital genetic data, can help to define not only the complete molecular signatures of the disease, but also the cascade of events that induce or maintain such signatures. For example, differential transcriptome analyses of pathological vs. healthy brain tissue allow for the examination of protein-coding genes, non-coding RNAs or splicing events that are differentially expressed in the two conditions [74,75,76,77]. Thanks to the combination of all these data, it is possible to provide an “omic” profile of the patients, discover networks and possibly contribute to better understand the neurological disease mechanisms at 360°. 

Indeed, the prospect of broadly applying a precision medicine approach has been dramatically improved by the recent development of large-scale biological databases, powerful methods for characterizing patients, such as proteomics [78], and computational tools for the analysis of large sets of data. It is therefore desirable that genome-wide association studies (GWAS) and NGS allow us to identify preclinical disease stages, formulate an adequate differential diagnosis and provide early and optimized therapeutic strategies to replace classical treatments [71]. 

Even though there is still much to understand, encouraging improvements in sequencing technologies has driven new and hopeful discoveries in the field of neurodegeneration. After cancer, in fact, neurological diseases offer the most compelling opportunities to achieve precision medicine with innovative approaches in which the health care of the individual is developed “tailored-based” on factors such as genetics and lifestyle [79]. 

From a historical point of view, precision medicine fully enters the world of neurodegenerative diseases thanks to the largest European network known as “Joint Program Neurodegenerative Disease Research” (JPND), active since 2011, which involves around 30 different countries aimed at tackling the challenge of neurodegenerative diseases—in particular AD.

This large network has launched a EUR 30 million call for proposals about precision medicine projects on neurodegenerative diseases, with the goal of finding out the causes and better treating those pathologies, such as AD and other forms of dementia, for which there are still no cures. Researchers are, therefore, working to also bring molecular medicine into the research of neurodegenerative diseases in order to make progress similar to those already made in oncology [80].

To make the potential of precision medicine in neurological disease realistic, however, it is necessary that many distinct areas of basic and translational research collaborate, in order to implement the precision medicine strategies across different specialized centers. The development of these networks will be able to facilitate the collection of data and to share the information derived from different studies. In this context, the involvement of a large number of samples from healthy individuals, who can provide control data to compare with those of patients, is also essential. By combining genetic information, clinics and patient images together, it will be possible to gradually solve the puzzle, which will lead to a clearer picture of the disease. 

In this regard, a great example of a web-based platform is represented by the “Italian IRCSS Network of Neuroscience and Neurorehabilitation”, whose purpose is to standardize and optimize patients’ clinical care and the therapeutic strategies applied to neurodegenerative diseases [81].

Another important objective, pursued by the European network, concerns the ability to standardize the way in which diagnoses are made, across international guidelines. Indeed, it is not enough for the diagnosis of a neurodegenerative disease to be correct, but until it is achieved identically in all European countries, it will not be possible to have sufficient cases of study with homogeneous data to catalogue each disease and its numerous subtypes. The aim is, therefore, to be able to find a common code that will help both research and patient.

Collectively, these findings could lead researchers to focus on rare variants in neurological diseases in parallel to the developments of new NGS technologies that facilitate the complete investigation of the human genome. 

As President Obama pointed out in the “Precision Medicine Initiative”, continuous efforts are necessary to transform information into knowledge so that neurological patients can receive better treatments by using their individual genome characteristics [29], as shown in Figure 3. 

Although the road is still long, recent advances in some neurodegenerative diseases testify the potential of precision medicine approaches in this field. 

Recent progress related to microRNA (miRNA) pharmaco-epigenomics, defined as the use of drugs to treat epigenetic defects connected to a disease, offers new strategies for the development of more effective treatments [82]. The implication of miRNAs in several brain disorders has, indeed, been extensively studied and many authors reported evidence of drug response regulated by miRNA-mediated mechanisms. In particular, in two independent studies, Alieva et al., and Margis et al., reported that the expression levels of some specific miRNAs (i.e., miR-7, miR-9-3p, miR-9-5p, miR-129 and miR-132) can be restored in patients affected by PD through the treatment of L-dopa, a dopamine receptor antagonist [83,84].

Additionally, concerning AD, although to date there is no cure, the treatment of patients with some drugs seems to improve or stabilize some common symptoms, such as memory deficit. About that, the use of specific drugs, such as cholinesterase inhibitors or dopamine agonists, seems to ameliorate memory deficits and inflammation via modulating the expression of miR-206 [85], known to target the BDNF (brain derived neurotrophic factor) transcript, a known neuroprotective factor against neuron cell death and various brain insults [86].

However, although these advances provide a more comprehensive picture of the complexity of neurodegenerative pathologies, the use of miRNAs as potential therapeutic targets still remains controversial and need to be further investigated, especially with regard to the methods of delivery and the target specificity.

Regarding personalized medicine in the field of neurodegeneration instead, a greater number of studies with encouraging results have been conducted. In fact, thanks to the great advances in sequencing technologies, it is now possible to obtain, quickly and at low cost, a large amount of information on the individual genetic profile to do prevention or direct targeted clinical studies. 

About that, WGS or WES approaches testify the importance of identifying novel rare genetic variants that led to a better understanding of the pathology and the development of new therapies. For example, concerning AD, the majority of variants in the sequence of the genome that seem to markedly affect the risk of disease are mostly present in three main genes encoding for APP (amyloid precursor protein), PSEN1 and PSEN2 (PRESENILIN 1, and PRESENILIN 2) [87]. However, although these variants appear to be fully penetrant in cases before the age of 60 years, they do not clarify the late-onset form of AD (LOAD) that seems to be instead strongly connected with rare variants. 

To mention some studies about this topic, Cruchaga and colleagues reported the discovery by exome-sequencing of a rare variant in PLD3, a gene codifying for a phospholipase highly expressed in brain regions that are vulnerable to AD, which seems to double the risk for that pathology in seven independent cases of LOAD [88]. 

With the whole-genome sequencing approach, Jonsson and colleagues identified a rare missense mutation (rs75932628-T) in TREM2 gene encoding for the triggering receptor expressed on myeloid cells 2, which confers a significant risk of AD in Icelanders [89].

Moreover, the recent advance of NGS technologies, coupled with the development of resources, such as the Human Genome Project and the International Human Haplotype Map Project, have provided the basis for genome-wide association studies (GWAS) in neurological diseases, as shown in Table 2.

The targeted sequencing approach of confirmed GWAS loci, such as ABCA7, BIN1, CD2AP, CLU, CR1, EPHA1, MS4A4A/MS4A6A and PICALM, identified an excess of rare deleterious mutations in three independent LOAD cohorts. The high coverage of sequencing allowed for the identification of these variants that could not have been detectable by WES or WGS [96].

Similar progress has also been obtained concerning precision medicine in the PD field.

GWAS have allowed the identification of multiple loci associated with PD48. However, in this case, the presence of rare variants within these loci may also contribute to an increase in the susceptibility to the disease. 

In 2019, Germer et al. used the WES approach in a German cohort to verify the presence of rare mutations in genetic loci, previously associated with PD by GWAS. The study evidenced the presence of 54 potential disease-relevant variants in 71 genes, suggesting that some of the associations identified could refer to rare variants with probable functional effects that modify the PD risk [97]. 

Interesting studies have been conducted about PD in Black South African and Nigerian patients, in which the disease seems to have quickly increased in recent years. However, a large number of PD cases still remains genetically unexplained, since no common mutations have been detected in the key genes responsible for PD in this population, while target sequencing permitted the identification of rare sequence variants. As a consequence, the limited number of variations in these patients suggests that the well-known PD genes may play a minor role in causing the disorder in these populations and that other genes are likely to be involved [98,99].

In Gialluisi et al., an exploratory WES analysis of 123 PD patients from Italy and an exome-wide association study of motor and non-motor PD phenotypes to test genetic associations with neurological disabilities were reported. The study evidenced a new variant associated with PD (rs201330591) in the GTF2H2 gene, previously implicated in spinal muscular atrophy (SMA), which was not replicated in other independent European cohorts [100,101]. 

Previously described examples of high-throughput sequencing of patient genomes are useful approaches of patient genomes for the sub-classification of phenotypically similar but genetically heterogeneous diseases, such as AD or PD, and allow for the identification of new causative alleles which could facilitate precise disease diagnosis and the treatment of rare diseases. 

### 4.2. Cardiovascular Diseases

Cardiovascular diseases consist of a range of diseases related to the circulatory system that could appear suddenly and unexpectedly and lead to death, or could represent the initial stage for a long-term condition that has a profound impact on healthcare costs. 

Despite many efforts and progress made towards the eradication of CVDs, such as modification of lifestyle, as well as evidence-based therapies, CVDs still remain the leading cause of death in western countries, accounting for ~32% of all global deaths. Unfortunately, the projection is even worse, as this trend seems to be increasing and it has been estimated that the number of deaths will rise to >23.6 million annually by 2030 [25]. These projections, together with an increment of global health burden and total cost of medical care, contributed to increasing research funding on CVDs.

There are different approaches and technologies aiming to treat or prevent CVDs. By way of the Human Genome Project (HGP), new strategies based on genome-wide predisposition markers, pharmacogenetics and genomic signatures have been developed. HGP opened the research to a new route to investigate cellular mechanisms, offering detailed information about the structure, organization and function of human genes.

Progress in CVD, providing novel insights into genetic architecture (genetics variants, frequency, magnitude of effect), derived from Genome-Wide Association Studies (GWAS), as shown in Table 3. These studies led to the identification of many loci associated with CVDs and risk factors [102,103]. Furthermore, GWAS identified novel pathways and improved treatments evaluating novel drug targets and developing personalized approaches for individual patients [104].

In cardiovascular medicine, the personalized medicine approach is an innovative tool that could be applied to all stages that characterize disease development, including risk prediction, preventative measures and targeted therapeutic approaches.

Among CVDs, coronary artery disease and dilated cardiomyopathy are of particular interest to personalized medicine approaches. In these conditions, different factors contribute to the onset of pathologies, as well as to a precise diagnosis of such conditions, ranging from genetics to epigenetics and proteomics.

### 4.3. Calcific Disease of Aortic Valve

The calcific disease of the aortic valve (CAVD) can be asymptomatic (aortic sclerosis) or bring about the remodeling of the valve tissue, leading to hemodynamic problems in the aortic valve, such as aortic stenosis (AS). The echocardiogram is currently the golden standard in CAVD diagnostics [117] but does not allow for the evolutionary prediction of the disease and, consequently, does not allow for the possibility of intervention in the early stages of the disease [118]. Computed tomography increasingly represents a useful tool for the classification and quantification of the calcification of the aortic valve [119] but it also only allows for a prediction on the severity and not on the evolution of the disease [120]. The role of biomarkers representing a predictive value in the development of cardiovascular events has been widely studied in cardiovascular atherosclerosis. Although some biomarkers have been identified as causal disease factors, the identification of their relationship with genetic factors in CAVD is still ongoing [118].

In a clinical study with a population of 5201 subjects over the age of 65, the authors identified the correlation between Lp (a) lipoprotein and low density cholesterol (LDL) with CAVD [121]. Another smaller clinical study (101 cases of CAVD) highlighted the existence of the relationship between high levels of Lp (a) > 48 mg/dl with a high risk of aortic stenosis [122]. These data confirm the results of a previous study, published in 2003, where there was a relationship between a SNP (single nucleotide polymorphisms) in the Lp (a) locus (rs10455872) and the calcification of the aortic valve. This same study also showed that elevated levels of genetic Lp (a) are associated with an increased risk of CAVD [123]. More recently, researchers reported that a genetically low determination of Lp (a) levels reduces the risk of CAVD by approximately 37% [124] and that patients with elevated plasma Lp (a) levels and phospholipids, oxidized in apolipoprotein B (OxPl-apoB), have a greater propensity for the development of aortic stenosis [121]. The role of genetics on valvular pathology is becoming increasingly evident. A relationship between NOTCH1 mutations and the severe calcification of the aortic valve has been identified in animal models [125]. The deregulation of the NOTCH1h in endothelial cells [126] or in the cell interstitium of the aortic valve [127] seems to promote the development of valve calcification. Although only a small number of genetic mutations have been identified as the causes of disease, it is hoped that the availability of new technologies will lead to the discovery of additional genes implicated in human valve disease [128]. In this context, more recent studies have focused on the clonal hematopoiesis of undetermined potential (CHIP), which is defined as the presence of an expanded somatic clone of blood cells in people without other hematological abnormalities, common among elderly people and associated with an increased risk of hematological cancer. CHIP has also been associated with a sharp increase in the risk of cardiovascular disease, such as myocardial infarction and atherosclerosis [129]. Mutations identified in the somatic blood cell clone most frequently influence four genes: DNMT3A; TET2; ASXL1; JAK2. [129]. Each mutation has been individually associated with coronary artery disease. In this study, the authors used a modified nested case-control study design from two prospective cohort studies: Malmö’s Diet and Cancer (MDC) Study, a community-based, prospective observational study of ~30,000 participants who were residents in Malmö (Sweden) [130]; BioImage Study, a multi-ethnic, observational study aimed at characterizing subclinical atherosclerosis in 6699 US adults [131]. CHIP carriers with these mutations also have an increase in coronary artery calcification, an indicator of coronary artery atherosclerosis load. The most characterized mutations are in TET2, a gene involved in epigenetic regulation. The mechanism by which mutated TET2 promotes the development of cardiovascular disease has not yet been fully understood; however, used as model in atherosclerosis-prone, low-density lipoprotein receptor-deficient (Ldlr−/−) mice, it has been shown that TET2 mutations favor the recruitment of macrophages in the endothelium, thus promoting the inflammatory state, which is believed to cause the rapid development of atherosclerotic plaques [132].

The pathogenesis of CAVD shares many similarities with the atherosclerosis process. Macrophage-mediated inflammation plays an important role in the calcification of the aortic valve [133], although Notch1 has been frequently associated with macrophage activation [134]. Recent data also show that the absence of HES1, the NOTCH target gene, improves the expression of cxcl1, a chemokine involved in the adhesion of macrophages to the endothelium [135]. Mutations induced by TET2 cause an increase in the expression of inflammation mediators, including cxcl142, and it has recently been observed that TET2 is able to epigenetically control NOTCH1 [136]. These data suggest that TET2 mutations in macrophages can lead to NOTCH deregulation and predispose the onset of aortic valve stenosis or promote its progression. It seems possible, therefore, that for the development of hematopoietic clones with somatic mutations, TET2 could, similar to what occurs in atherosclerosis, constitute a risk factor for the development of aortic valve stenosis.

### 4.4. Dilated Cardiomyopathy

Dilated cardiomyopathy (DCM) is defined by the observation of progressive dilation and the deterioration of the contractile function of the left ventricle in the absence of obvious causes of pressure and/or volumetric overload, such as the presence of chronic ischemia, hypertension and/or diseases acquired or congenital valves [137]. The causes of this pathology are numerous and heterogeneous and, although the etiology of DCM is commonly divided into genetic and non-genetic causes, many of those DCM diagnosed do not find a precise etiological definition. Another important aspect of DCM is the enormous variability in clinical presentation in terms of both severity and age of onset and prognosis. A positive family history is evident between 30–50% DCM, while a causal gene can be identified in up to about 40% of cases [138]. Genetically determined alterations of the sarcomere, mitochondrial alterations and neuromuscular pathologies are among the most frequently identified etiologies in the presence of a family history, while acquired diseases or environmental conditions, such as exposure to toxic substances, pregnancy, diabetes or myocarditis, contribute to the appearance of the phenotype and prognosis. Conversely, a genetic alteration, not yet identified, can increase susceptibility to environmental factors, favoring the appearance of DCM with a highly variable phenotype. In DCMs defined as “idiopathic”, the condition of familiarity is found in about 30% of cases [138]. However, traditional gene sequencing strategies allow fewer genes to be identified candidates, mainly involved in autosomal dominant forms and with variable penetrance. Consequently, in most patients with DCM, the etiopathogenesis remains almost indefinite, with consequent difficulties in the therapeutic approach and in the prognostic definition. How can precision medicine improve the diagnosis of these pathologies to guide clinicians toward the more personalized treatment of patients? The advent of next-generation sequencing technology (NGS: Next Generation Sequencing) added a new dimension in genomics research thanks to the generation of data with massive and ultrafast methods, allowing for more extensive genome coverage. This approach has led to the identification of an increasing number of genes and mutations responsible for DCM. This genetic variability is probably related to the extreme heterogeneity of disease manifestation. Important findings have associated mutations of LAMININ A/C (LMNA) and FILAMIN C (FLNC) to poor prognoses and the propensity to cause an arrhythmic phenotype, respectively, while TNNT2 mutations have been associated with early onset aggressive disease [139,140,141,142,143].

The importance of recurring genetic tests is still a matter of debate; however, according to guidelines [144], the genetic tests could be helpful in case of doubtful diagnoses or borderline cases, otherwise they are extremely important in cases of family history or in the presence of other risk factors. For example, in patients with a family history of pacemaker implantation or sudden cardiac death (SCD), or evidence of atrioventricular or intraventricular conduction delay, the presence of the LMNA mutation could represent an alarm bell, leading to the decision of an early ICD implantation [145].

In 2015, Lee and Ware [146] demonstrated how the genetic diagnosis of pediatric cardiomyopathy influences patient management. They reported, as an example, three case studies that documented the benefits of genetics tests. In the first case, genetics helped to diagnose Nanoon syndrome, associated with HCM. This was extremely important for patient management, since Nanoon syndrome is associated with valvular issues which may require balloon valvuloplasty, surgery or other intervention. In a second case, genetic tests revealed a sarcomeric mutation in an 18-year-old male with HCM. After this result, his brother was subjected to examinations and mutation-specific genetic testing. In this case, genetic tests helped clinicians to formulate more specific recommendations as well as to identify other family members who may be at risk. The third case reported a six-year-old female, referred for heart transplant evaluation. This young girl was subject to genetic testing for a glycogen storage disease suspicion. This case demonstrates the etiological heterogeneity that exists, especially within pediatric cardiomyopathy, where DCM could be related to genetic disease rather than myocarditis.

Additionally, on the high-resolution imaging front, several technological advances have led to the characterization of cardiac remodeling mechanisms, as well as specific alterations, with more specificity structural effects of the myocardium and/or the type of heart damage.

The correct etiological definition has several consequences both from the knowledge point of view, as well as pathophysiological, therapeutic and prognostic points of view. From the pathophysiological point of view, for example, some genetically determined alterations of mitochondrial functions may increase the possibility of the development of DCM in the presence of specific environmental exposures or certain therapies. The mitochondrion is fundamental for the continuous request of energy in the form of ATP by the myocardium to carry out its contractile functions. The alteration of the mitochondrial functions determines not only a reduction in ATP production but also an imbalance in the redox system, favoring the production of free radicals of O2. Such alterations are described, for example, in alcoholics but also in subjects subjected to specific chemotherapy drugs. Hence, genetically determinant alterations of the mitochondrion, in silent genera, give rise to a DCM phenotype in the presence of such conditions. Still, the therapy of DCM does not include a specific guideline, but is based on the classic therapy planned for affected patients of chronic heart failure, where other causes, such as hypertension, diabetes or ischemic heart disease, are prevalent. The ability to define the etiopathogenetic characteristics of DCM provides essential support for the development of new strategies, pharmacological and otherwise. Finally, the integration of genetic data with instrumental data (MRI and echocardium) can allow for finely defining the patient prognosis, identifying potentially reversible structural changes after the elimination of environmental exposure or toxic factors.

The genetic tests in patients with cardiomyopathy present several benefits. In the case of doubtful diagnoses, such as unexplained left ventricular hypertrophy, positive genetic testing could guide clinicians towards appropriate screening and medical therapy and recommendations about physical activity, as well as family surveillance. These tests could also identify the etiology of pathology allowing for appropriate cardiac screening and medical therapy and the appropriate management and surveillance of other organ system involvement. Finally, they are extremely useful for identifying at-risk relatives, improving risk stratification and the cost-effective implementation of cardiac surveillance only in at-risk individuals [146].

## 5. Conclusions

In recent years, personalized medicine has become a hot topic in many fields of research and it will probably become of major importance in the future. The big interest in this area could be explained by the development of biology systems and high-throughput technologies. Certainly, the increasing knowledge and interpretation of data from genetic analyses will enhance our understanding of physiological events during health and disease and promote personalized diagnosis and treatment. This kind of approach could also be beneficial to reduce the burden of disease by targeting prevention and treatment more effectively through the integration of inputs from multiple data sources. Moreover, personalized medicine also aims to decrease healthcare costs and prevent adverse events by improving the ability to select the right therapy at the right time for an individual patient.

However, personalization cannot be extended to all pathologies. In fact, different diseases need to integrate biological, non-biological and environmental factors in decision-making. In complex diseases, peripheral components of the biological network would be more likely to contribute to the disease than genetic determinants, which are also more probabilistic than deterministic for the development of the disease.

Moreover, the significance of any specific approach to personalized medicine still needs to be tested. Theoretically, any proposed personalized approach must be confirmed by clinical trials to demonstrate benefit above current guideline-based therapy.

One important feature of the personalized medicine approach involves the patient and his/her participation as a central and active actor of the complex network. So, each individual being aware of their medical history and first phase education and training for healthcare providers and patients are essential to prepare for the future where a personalized medicine approach could be realized.

Furthermore, an important issue concerning the results of genomics testing is “data privacy” management [147]. We have to discern “research” from “clinical” data privacy. The development of genomic data led to a rapid growth in the collection of bio-samples and the creation of biobanks. In the US, as well as in Western Europe and others countries, different programs and legislation regulate the handling of sensitive data to protect the privacy of individual participant studies by keeping genetic data private, as stipulated in informed consent agreements [148]. From the clinical point of view a relevant issue derived from the results of genomic testing is the “return” of these results to patients, mainly in the case of “incidental findings”. The matter is what is “right to know and what is right to not know.” Not all patients want to know or can understand the meaning of genetic results, mainly when they predict the risk of the development of cancer or other chronic disease. This could cause anxiety or depression in patients and their families and, in some cases, this could results in unnecessary harm, since not all genetic susceptibility can be expected to develop into the disease in the future [9,149]. On the other hand, if the individual or his/her siblings develop the disease that could be “treated in time” in light of genomic results, this could be considered as an ethical injustice. There are only a few guidelines or recommendations addressing this issue and they often differ among countries. This has become a relevant problem in the case of international collaborations where scientists need to share data and samples. One possibility to overcome this question is to face it during the planning and design of projects before submission to Ethics Committee Board for approval. The informed consent module should contain clauses concerning this issue. Nevertheless, the physician who administers the informed consent for the agreement must be sure to clarify the risks and the benefits to participants of the study and ensure that the patients are prepared for the eventual incidental findings before enrolling them in the study. 

Personalized or precision medicine aims to customize medical practice with a focus on the individual, based on the use of genetic tests, the identification of biomarkers and the development of targeted drugs. From this perspective, public healthcare systems play a key role, with a wealth of new possibilities promised for patients and society through the increased adoption of personalized approaches to medicine. This has significantly impacted the treatment of some types of chronic diseases, with a direct impact on the cost of public health and individual health care. 

Thus far, many barriers still exist and the promises to achieve personalized or precision medicine in various specialty fields and its effect on public health and medical education, still remain unfulfilled. The greatest challenge is to integrate personalized medicine into contemporary medicine [150]. Prithcard et al. in 2017 reported the results of a survey conducted by the Healthcare Working Group (HWG) of Personalized Medicine Coalition (PMC) [151]. PMC’s HWG identified five areas of challenges: education and awareness; patient empowerment; value recognition; infrastructure and information management; ensuring access to care. They also discussed strategies for overcoming these challenges, making a step towards to the evolution of healthcare delivery to personalized medicine 

Little is known about the impact of socioeconomic health inequalities. In fact, the high cost of biotechnologies can amplify health inequalities and become a problem for health service sustainability, especially for populations who stand to benefit the most from these technologies but are unable to afford, access or use them [152].

## Figures and Tables

**Figure 1 genes-11-00747-f001:**
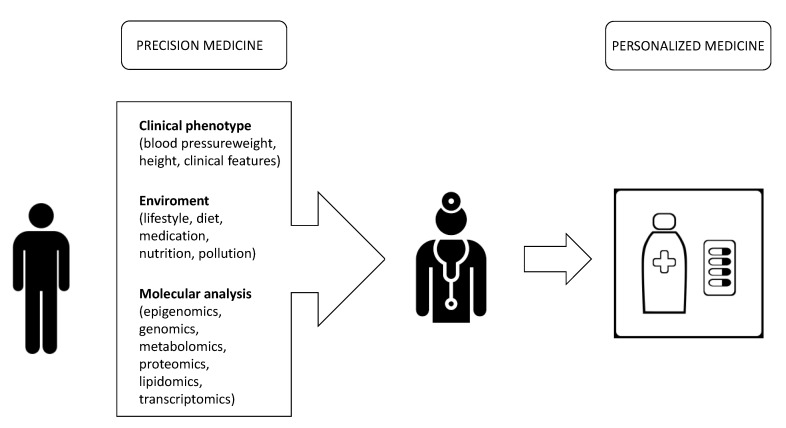
An ambitious challenge for medicine is to guarantee targeted care paths, beginning with more personalized approaches. To achieve this goal, it is necessary to have a multi-level approach towards patients. At molecular level the multi omics approach (transcriptomic, metabolomics, genomic, proteomics, epigonomics) provides a deeper understanding of patient conditions from the original causes of diseases to the functional consequences. This information should be integrated with the study of the “exposome”, defined as the totality of exposure experienced by an individual during their life and the health impact of those exposures (Wild CP. Complementing the genome with an “exposome”: The outstanding challenge of environmental exposure measurement in molecular epidemiology. Cancer Epidemiol Biomarkers Prev. 2005; 14(8):1847–1850). Together with the study of clinical features of patients, physicians are able to elaborate a personalized therapy, tailored to the individual patient.

**Figure 2 genes-11-00747-f002:**
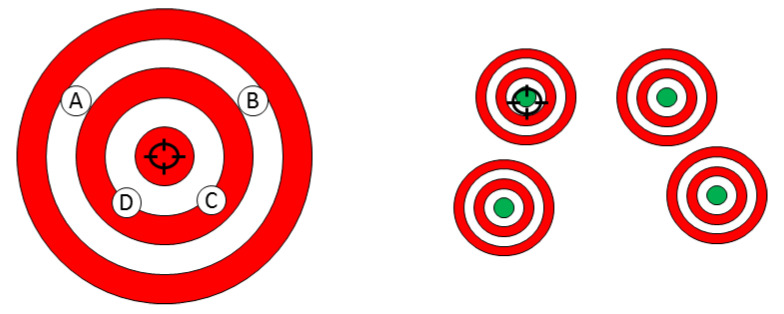
Standard approach assumes that all patients with the same symptoms of disease share a common patho-phenotype and, therefore, should be treated similarly. To reach the goal, that is, the recovery of patients, physicians have at their disposal different therapies (here indicated as A, B, C, D) that they have to “test” on patients until they find the right one. Conversely, personalized medicine aims to improve the ability to select the right therapy at the right time for an individual patient.

**Figure 3 genes-11-00747-f003:**
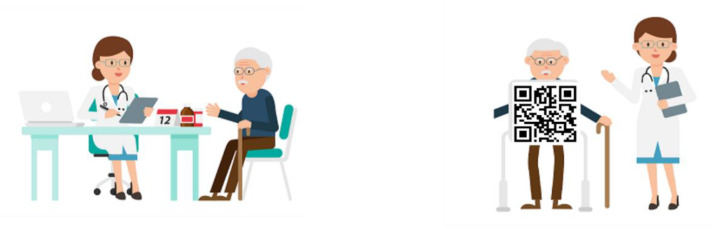
As noted by President Obama in “Precision Medicine Initiative”, the use of individual genome characteristics is essential to choose the best and adequate treatments for patients. Modified from: Doctor with Patient Cartoon.svg and Doctor with Patient X-ray Cartoon.svg from Wikimedia Commons by Videoplasty.com, CC-BY-SA 4′′.

**Table 1 genes-11-00747-t001:** Principal applications, advantages and limitations of different NGS strategies.

Technique	Target Regions	Variants Detected	Advantages	Limitations
WGS	Entire genome	~4,000,000	Identifies variants in all genome Detects genome rearrangements and structural variants Uniform depth of sequencing	Highest cost Largest volume of data is produced Require long and most complex analysis Limited application in clinical diagnostic
WES	2% of genome	~20,000	Identifies variants in all protein-coding regions Low cost compared to WGS	Possibility to have incomplete exome coverage Cannot detect non-coding and structural variants Require exome capture or enrichment methods during library preparation
Target panels	Few genes	Variable: depends on the panel size	Identifies variants in specific regions Customizable Lowest cost Rapid analysis Short running time More suitable for clinical applications	Variants limited to selected genes Limits in the identification of novel variants and structural variants Needs continuous updates as a result of new discoveries

**Table 2 genes-11-00747-t002:** Review of published GWAS. The table summarizes susceptible genes in Alzheimer’s and Parkinson’s disease identified by GWAS. Genes replicated across different studies are shown in blue.

Gene/Locus	Disease	Reference
TREM2	Alzheimer’s	Jonsson et al., 2013 [88]
ABCA7, BIN1, CD2AP, CLU, CR1, EPHA1, MS4A4A/MS4A6A, PICALM	Alzheimer’s	Vardarajan et al., 2015 [90]
KIF5A	Alzheimer’s	Nicolas et al., 2018 [91]
EXOC3L4	Alzheimer’s	Miller et al., 2018 [92]
PSMF1, PTPN21, ABCA7, ACE, EPHA1, SORL1	Alzheimer’s	Zhao et al., 2019 [93]
DNAJB2, HSJ1	Parkinson’s	Sanchez et al., 2016 [94]
22q11.2	Parkinson’s	Butcher et al., 2017 [95]
PRKN	Parkinson’s	Bravo et al., 2018 [90]
DNAH1, STAB1, ANK2, SH3GL2, NOD2	Parkinson’s	Germer et al., 2019 [96]

**Table 3 genes-11-00747-t003:** Review of published GWAS in cardiovascular disease.

Genes	Associated CVD	Reference
PCSK9	Myocardial Infarction	Myocardial Infarction Genetics Consortium (2009) [105]; Abifadel M. et al. (2003) [106]; Cohen J.C. et al. (2006) [107]; Teslovich T.M. et al. (2010) [108]
PDGFD	Coronary Artery Disease.	Coronary Artery Disease C4D Genetics Consortium (2011) [109]
LRIG3	Congestive Heart Failure	Smith N.L. et al. (2010) [110]
ZBTB17 BAG 3 MYBPC3 LMNA PLN	DCM	Villard E. et al. (2011) [111]; Walsh R. et al. (2017) [112]; Hershberger R.E. et al. (2013) [113]
PITX2 SCN10A	Atrial Fibrillation	Jabbari J. et al. (2016) [114], Roselli C. et al. (2018) [115]
ACTN2	Hypertrophic Cardiomyopathy	Chiu C. et al. (2010) [116]

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
