# Peer review of "Precision and Personalized Medicine: How Genomic Approach Improves the Management of Cardiovascular and Neurodegenerative Disease"

_genes, 2020, doi:10.3390/genes11070747_

Round 1
Reviewer 1 Report
In this manuscript, authors have revised “the state of the art “of the precision medicine for cardiovascular and neurodegenerative disease. They describe the genomic technologies available and show samples of its application to these diseases.
The manuscript needs to be improved and some considerations are provided below.
- The title is too general, across the manuscript readers discover that authors only focus the information in several pathologies of each disease, for example, Parkinsonism and Alzheimer´s diseases. Other neurodegenerative diseases could be taking into account.
- To clarify figure 1 letters A, B, C, D should be added to the small target.
- Line 162 I´d like to remember to the authors that Sanger sequencing is already the “gold standard” of the sequencing in clinical practise. Maybe, authors should mention this data.
- Line 165, nowaday, Third generation sequencing (3rdGS) is the term that it is used to designate the last sequencing technology and it should be included.
- Line 166 Next Generation Sequencing (NGS) need to be added
- Proteomic and metabolomic approaches are also important in Precision medicine, then authors should include these approaches too.
- Lines 196-199. I don't agree with this affirmation. Nowadays, it has been described methods that allow the identification of very low-frequency variants.
- Line 219, the Authors should have into account that Digital PCR only allows the identification of a single variation. In the case of two variants present in the same amplicon the Digital PCR is not able to detect both of them.
- Line 252. “the most comprehensive” There are a lot of custom and commercial panel to study cardiomyopathies that are as useful as Trusight cardio panel. Then, maybe this phrase should be changed.
- Lines 261 272. Nowadays The TGS technology shows the limitation that has a large error rate and its use in clinical practice is quite limited.
- Line 322 maybe the word “digital” should be changed.
- Gene names must be in cursive font.
- Line 438. To support the affirmation “was not replicated in other independent European cohorts” a reference need to be added
- line 439 to 442 This paragraph is not clear, please rephrase it
- Figure 3 doesn't provide any extra information.
- Maybe a review of the GWAS studies published could be shown as a table.
- Line 495and line 509-510 NOTCH and TET2 genes must be in capital letter
- Line 506 Maybe, a reference for each gen could be added.
- The paragraph from line 509 to 513. For readers, it is not clear if there are any mutations in TET2 associated with CAVD or there are only studies in mouse. Please clarify it.
- The paragraph from line514 to 524 it is not clear if these data were obtained in humans or mice
Author Response
We thank the reviewer for all comments. We tried to best address our aim and modified the manuscript according to his suggestion
Point 1 The title is too general, across the manuscript readers discover that authors only focus the information in several pathologies of each disease, for example, Parkinsonism and Alzheimer´s diseases. Other neurodegenerative diseases could be taking into account.
Response 1
We appreciated this suggestion. We changed the title with a more appropriate one. We also reorganized the introduction to address the reader toward the right aim of this review
Point 2 To clarify figure 1 letters A, B, C, D should be added to the small target.
Response 2
We clarified the figure as requested by the reviewer
Point 3 Line 162 I´d like to remember to the authors that Sanger sequencing is already the “gold standard” of the sequencing in clinical practice. Maybe, authors should mention this data.
Response3
We clarified that the Sanger sequencing is the gold standard in diagnostic procedures (lines 181,82)
Point 4 Line 165, nowaday, Third generation sequencing (3rdGS) is the term that it is used to designate the last sequencing technology and it should be included.
Response 4
In line 165 we wanted to underline the transition from First Generation Sequencing (Sanger Method) to Second Generation Sequencing also defined Next Generation Sequencing. In the review, we also introduced the new sequencing technologies, that in the old version of the text, were called TGS, but in the new version, to be clearer, have been defined as Third Generation Sequencing (3rdGS), lines 279,280, 282, 284.
Point 5 Line 166 Next Generation Sequencing (NGS) need to be added
Response 5
We added the expanded word, Next Generation Sequencing (NGS), in line 187
Point 6 Proteomic and metabolomic approaches are also important in Precision medicine, then authors should include these approaches too.
Response 6
We thank the reviewer for this suggestion. We agree that metabolic and proteomic are two relevant approaches, however, the aim of this review is to show how the genomics approach is useful for the diagnosis and management of cardiovascular and degenerative diseases. Otherwise, the title was too general and didn’t address the reader to the right aim of this review, we provided to change the title choosing for a more tailored one
Point 7 Lines 196-199. I don't agree with this affirmation. Nowadays, it has been described methods that allow the identification of very low-frequency variants.
Response 7
We agree with the reviewer, today there are several methods that allow us to identify rare variants. What we wanted to say in the text is that NGS methods allow studying variants with a low frequency, not in a single gene or target position, but extending the variants identifications to a wider scale, reaching also the whole genome evaluation, also when low DNA amounts are available. To better clarify this point we revised the sentence (lines 218).
Point 8 Line 219, the Authors should have into account that Digital PCR only allows the identification of a single variation. In the case of two variants present in the same amplicon, the Digital PCR is not able to detect both of them.
Response 8
We added this characteristic of Digital PCR in the revised text (lines 240-242)
Point 9 Line 252. “the most comprehensive” There are a lot of custom and commercial panel to study cardiomyopathies that are as useful as Trusight cardio panel. Then, maybe this phrase should be changed.
Response 9
We corrected this information in the text (lines 276)
Point 10 Lines 261 272. Nowadays The TGS technology shows the limitation that has a large error rate and its use in clinical practice is quite limited.
Response 10
We thank the R. for the observation made. We added in the revised text the limitations associated to TGS technology (lines 297-299)
Point 11 Line 322 maybe the word “digital” should be changed.
Response 11
We changed it
Point 12 Gene names must be in cursive font.
Response 12
We fixed it
Point 13 Line 438. To support the affirmation “was not replicated in other independent European cohorts” a reference need to be added
Response 13
We added the reference to the revised manuscript.
Point 14 line 439 to 442 This paragraph is not clear, please rephrase it
Response 14
We are sorry for the lack of clarity of this paragraph, we have rephrased it.
Point 15 Figure 3 doesn't provide any extra information.
Response 15
We eliminated it
Point 16 Maybe a review of the GWAS studies published could be shown as a table.
Response 16
We provided to include 2 tables summarizing main studies GWAS in Neurodegenerative and cardiovascular diseases
Point 17 Line 495and line 509-510 NOTCH and TET2 genes must be in capital letter
Response 17
We fixed it
Point 18 Line 506 Maybe, a reference for each gen could be added.
Response 18
We added new reference
Point 19 The paragraph from line 509 to 513. For readers, it is not clear if there are any mutations in TET2 associated with CAVD or there are only studies in mouse. Please clarify it.
Response 19
We clarified this paragraph (lines 550-555)
Point 20 The paragraph from line514 to 524 it is not clear if these data were obtained in
humans or mice
Response 20
We better described these studies
Reviewer 2 Report
It would be good to include more figures and tables.
Author Response
We appreciated the comment of the reviewer
Point 1 It would be good to include more figures and tables.
Response 1
As requested by the R. we have included 3 new tables, Table 1, including the principal applications, advantages and limitations of different NGS strategies and T 2,3, Review of published GWAS. We also included a new figure
Reviewer 3 Report
Dear Authors,
The authors Oriana Strianese et al. have explained the importance of Precision medicine in treating Cardiovascular disease and neurodegenerative disease in this review. This manuscript is predominantly written clearly and fits the scope of the journal. In this review, the author clearly explained about precision medicine and its latest development. Here are my comments
- Precision medicine is all the Panomics (Genomics, transcriptomics, epigenomics, metabolomics, proteomics, and microbiomes) taken into consideration to treat an individual. The author has failed to explain the omics and importance of the GWAS project in this review.
- Circ Res. 2018 Apr 27; 122(9): 1302–1315. In this review, the authors clearly explain the precision medicine in CVD. Based on this, The author needs to justify the novelty and importance of this review.
- The author should consider using more pictures to convey their messages.
- The title and introduction part does not correlate. The author has emphasized more on the difference between precision medicine and personalized medicine too much. Best way to explain it by giving examples.
- The author also needs to address important factors which determine the future of precision medicine – Finance or Health care provided or Medicinal practice, which ones play an important role.
- The author also needs to discuss data privacy
Author Response
Thank the reviewer for all comments. We provided a new version of manuscript addressing the issue suggested by the reviewer
Point 1 The author has failed to explain the omics and importance of the GWAS project in this review.
Response 1
Thank to reviewer for this comment. We included 2 tables summarizing the main findings of GWAS in cardiovascular and neurodegenerative diseases
Point 2 Circ Res. 2018 Apr 27; 122(9): 1302–1315. In this review, the authors clearly explain the precision medicine in CVD. Based on this, The author needs to justify the novelty and importance of this review.
Response 2
Thank the reviewer for this comment. The paper of Leopold and Loscalzo inspired the idea to write the present review. Our work is quite different from the above mentioned. In the Circ Res paper the authors emphasized the importance of phenotype as a core principle of precision medicine. In this review our aim is to better describe the differences between precision and personalzed medicine giving an example of their application in clinical medicine.
Point 3 The author should consider using more pictures to convey their messages.
Response 3
We added a new figure and 3 new tables
Point 4 The title and introduction part does not correlate. The author has emphasized more on the difference between precision medicine and personalized medicine too much. Best way
to explain it by giving examples.
Response 4
We changed the title of review with a more appropriate one
Point 5 The author also needs to address important factors which determine the future of precision medicine – Finance or Health care provided or Medicinal practice, which ones play an important role.
Response 5
Thank to reviewer for this suggestion. We addressed this issue in the conclusions (lines 699-711)
Point 6 The author also needs to discuss data privacy
Response 6
We discussed about this relevant question in the conclusions (lines 677-698)
Round 2
Reviewer 1 Report
the manuscript is now ready to be published
Reviewer 3 Report
NO comments